# Application of the TDR Soil Moisture Sensor for Terramechanical Research [note 1]

**DOI:** 10.3390/s19092116

**Published:** 2019-05-07

**Authors:** Jarosław Pytka, Piotr Budzyński, Mariusz Kamiński, Tomasz Łyszczyk, Jerzy Józwik

**Affiliations:** Faculty of Mechanical Engineering, Lublin University of Technology, Nadbystrzycka 36, 20-618 Lublin, Poland; p.budzynski@pollub.pl (P.B.); mariusz.kaminski@pollub.pl (M.K.); tomasz.lyszczyk@gmail.com (T.Ł.); j.jozwik@pollub.pl (J.J.)

**Keywords:** TDR sensor, soil moisture, terramechanics, wheel-soil interaction, vehicle mobility, snow density, grassy airfields.

## Abstract

This paper presents examples of the application of the TDR (Time-Domain Reflectometry) sensor in terramechanical research. Examples include the determination of soil moisture content during off-road vehicle mobility tests, the determination of snow density before and after the wheeling of a snow grooming machine and an airplane, as well as the monitoring of turf moisture on a grassy airfield for the analysis and prediction of safe and efficient flight operations (takeoff and landing). A handheld TDR meter was used in these experiments. Soil moisture data were correlated with the vehicle mobility index and a simple model for this correlation was derived. Using grassy airfield research, soil moisture data were related to meteorological impacts (precipitation, sunlight, etc.). Generally, it was concluded that the TDR meter, in its handheld version, was a useful tool in the performed research, but a field sensor that operates autonomically would be an optimal solution for the subject applications.

## 1. Introduction

Terramechanics is a science that deals with interactions between running gears of vehicles or machines and the soil or other soft, deformable surfaces, such as snow, grass, etc. The basic problem in terramechanics is analyzing and modeling the effects of soil conditions on wheel or track performance [1,2,3], especially the effect of soil moisture content. Soil, being a porous, three-phase, and highly hygroscopic material, changes its moisture content in response to atmospheric impacts, such as precipitation, solar radiation, wind, and ambient air temperature, as well as in response to the presence of vegetation. The dynamics of soil moisture can be very intensive and may significantly change within hours so it affects the mechanical properties of the soil material [4,5,6].

In terramechanics, some synonym terms are used to describe the performance of a vehicle on soft, deformable terrain. Soil trafficability is a set of soil mechanical properties that affect the generation of tractive forces, net traction, *T* (N), rolling resistance, *RR* (N), and drawbar pull, *DBP* (N) [7]. Mobility is a more general term, used widely in the automotive or transportation community to characterize the ability of being “on the move”, sometimes used interchangeably with vehicle dynamics [8,9]. However, in the terramechanics, the term mobility describes a situation when a vehicle passes over unpaved, natural terrain with a supply of traction, typically quantified by means of the drawbar pull, which can be expressed using the equation below:(1)DBP=T−RR.

But terramechanics also deals with other aspects of off-road locomotion, for example the effects of wheeling on soil compaction or safety and ride comfort. For most natural materials, like soils or snow, moisture content is an important physical property. Modeling the effects of soil moisture on soil trafficability, wheel performance, and vehicle mobility is a challenge and needs validation with data measured in real-field conditions.

One interesting terramechanical problem is wheel performance on natural grassy surfaces. Traction on wet grassland is considerably weakened and the resulting mobility, or vehicle dynamics, is lower when compared to a hard, dry surface. Not only does the braking force performance become lower due to the slippery grass, but also the bearing capacity of the soil underneath alters. Consequently, wheel sinkage and rolling resistance increase, and this situation is especially dangerous for airplanes operating on grassy runways [10,11,12].

This paper presents researches in which soil (or snow) moisture has been determined with the use of a handheld soil moisture meter together with vehicle mobility data or atmospheric conditions data for wheel-soil interaction modeling.

## 2. Materials and Methods

### 2.1. Off-Road Vehicle Mobility Tests

In this study, we aimed to perform full season mobility tests with the use of a real vehicle together with simultaneous soil moisture measurements.

Mobility tests were performed from April 2006 to March 2007 (full calendar year). The tests were performed once a week, typically on a Wednesday or Thursday, in the morning hours (9:00–11:00). The format of the tests was that a test vehicle drove a test track of 500 m in length and the following components were measured or observed:time to reach *v* = 30 km/h, *t_V_*_30_;time to pass the test distance (approx. 500 m), *t*_500_;absolute mobility in the sense of “go-no go”, *GNG* (value = 1 for “go”, value = 0 for “no go”);a need to use the 4 × 4 drive, AWD—All Wheel Drive (value = 1 for those tests where there was no need to switch to 4 × 4 mode, and value = 0, when the 4 × 4 drive was required to pass the test distance).

The test track was partially slopped (ca. 70 m at 4% slope) and partially horizontal (ca. 430 m), with three curves. The surface was a mix of loess and sandy soils with some gravel, compacted by minor traffic, since it was used as a provisional road. An approximate composition was: loess soil 70%, sand 20%, and gravel 10%. Original soil was loess, sand and gravel had been added in order to improve traction and increase vehicle mobility.

The tests vehicle used was the Suzuki Vitara sport utility vehicle (SUV) with a manually controlled (switch on/off) 4 × 4 drive. In the 2 × 4 mode, the rear axle of the vehicle was steering. Only one person (driver) was always on board during the tests. The tires used were Wrangler 205/75R15, M+S (mud+snow), off-road treaded. The test vehicle is shown in Figure 1.

The following formula was developed in order to quantify the mobility of the vehicle by means of a term, *VMI*, called the “vehicle mobility index”, from the measured or indicated components:(2)$$VMI=\frac{1}{5}\;\left(\frac{t_{v30}}{t_{v30}^{w}}+\frac{t_{500}}{t_{500}^{w}}+2GNG+AWD\right)$$
where: $t_{v30}^{w}$ and $t_{500}^{w}$ are the reference times obtained on a rigid (asphalt), dry surface.

Note the *VMI* values were within the range from 0 to 1. The formula given in Equation (2) respected the following effects upon mobility:*t_v_*_30_, longitudinal dynamics of the vehicle motion, in terms of the time of vehicle acceleration, from stop to reaching a velocity of 30 km/h, related to the dry asphalt conditions;*t*_500_, longitudinal dynamics of the vehicle motion, in terms of the time to drive the distance of 500 m, related to the dry asphalt conditions;*GNG*, the absolute mobility, quantified by means of the number of “go” cases observing during the tests; and*AWD*, the number of cases when the 4 × 4 mode was not required to pass the test.

One of the main goals of this part of the study was to determine the total effect of meteorological factors, such as precipitation, wind, sun radiation, and temperature, but without identifying or measuring them. The only variable was the calendar date of the day of mobility tests. From the aspect of wheel-soil interactions, it would be practical that all of the above mentioned individual factors would be taken together in a form of a single parameter or physical quantity. It was assumed that soil moisture content could be such a quantity in this case.

Soil moisture content was measured with the use of a handheld TDR (time-domain reflectometry) meter, originally developed by the Easy TEST/Institute of Agrophysics, Polish Academy of Sciences in Lublin, Poland. Four experimental sites, filled with four different soils, were tested: loess, sand, forest soil, and turf with grass vegetation. The sites were approx. 1.0 m × 1.0 m and the measurements were made during the entire 2006 season, with the exception of days with freezing conditions. Since the soil used for the mobility test track was mostly loess, the results of loess moisture content were taken for the analysis. Results of the three remaining soil were recorded for a reference in future research.

### 2.2. Monitoring of Turf Moisture Content of a Grassy Airfield

The dynamics of soil moisture on a grassy runway significantly influenced both the ground performance of the aircraft and the organization of flight operations. Due to the lack of information about the conditions on the grassy runway, it was not known when to stop flights and when they could be resumed. This les to dangerous situations (too “courageous” traffic management at the airport) or to long-term cancellation of flights.

Conditions on a grassy runway may change in less than 1 h. There are known situations when the conditions at the destination grassy airfield have changed during the flight so that the landing became dangerous. It also happens that the grassy runway at a transit airport, as a result of hydration after convectional rain, may not be suitable for aircraft take off to continue the trip. Taking into account the described situations, an analysis of the performed humidity measurements were carried out in terms of dynamics.

Usually in the spring months in Poland there is the highest dynamics of soil moisture, which are associated with intense convective precipitation, as well as high air temperature and solar radiation. In addition, the still small, growing vegetation contributes to both intensive irrigation, as a result of precipitation (less water intake through vegetation), and also rapid evaporation of water due to the action of wind. Therefore, for the further analysis, the period from 15 to 30 May 2014 was selected and adopted as the time of occurrence of high dynamics of soil moisture on a grassy runway.

Soil moisture content was measured on a grassy runway of the Radawiec airfield, located near Lublin, southeast Poland using a handheld TDR meter, as shown in Figure 2. This place was different from the experimental site of vehicle mobility tests. In the selected period, soil moisture measurements were carried out on a research plot covered with grass turf. The time interval between measurements was 3 h, but no measurements were taken when the weather situation was stable. Simultaneously, weather observations were carried out: air temperature, and wind and cloud coverage were measured. For the analysis, a few characteristic time sections were selected, in which intense factors affecting the dynamics of soil moisture (convective precipitation, high air temperature, strong wind) were observed.

### 2.3. Determining the Snow Density by Means of Snow Moisture Measurements

One unconventional use of the TDR meter is the possibility to determine snow density. Measurements were done by the authors in order to determine the initial snow conditions during measurements of snow stress by a grooming machine and its effects upon snow compaction at a ski resort [13], as well to compare wheeling versus skidding on the snow surface for a light airplane [14].

The experiment with a grooming machine was performed in March 2009 in a mountainous area at an altitude of approximately 1700 m above sea level. The region is extensively used by alpine skiers, inside Tatra National Park. On the day of the tests, the sky was 8/8 clouded and snow was falling and the air temperature was −6 °C. The temperature of the snow where the tests took place was −1.9 °C, its depth was 170 cm, and there was a layer of 7–10 cm of fresh fallen snow. A Bombardier snow grooming machine was used in this experiment. The vehicle was driven at a low speed of about 3–5 km/h. The experiment consisted of six passes forward and backward to obtain data for describing the effect of multiple passes upon snow stresses and the results are reported in [15]. Measurements with the use of the TDR meter were done in snow before the pass of the machine, at four depths: 10, 20, 30, and 40 cm.

In the second experiment, a four-passenger, short landing and takeoff (STOL) airplane was used. The airplane was equipped with a combined ski and wheel landing gear system, using the aircraft skis of 1.75 m × 0.50 m (length × width) that were mounted to wheel axles (see Figure 3 and Figure 4). A pneumatic system enabled the remote raising or lowering of the skis. Moreover, a small 0.75 m × 0.2 m ski was installed along with the tail wheel. The TDR snow measurements were performed together with snow stress measurements, which were done with the use of a pressure cell installed in the snow at 10 cm depth. The TDR probe was inserted into fresh snow as well as into the snow in the rut formed by the wheel and by the ski.

In both experiments, snow density was determined based on measured values of electrical permittivity of snow and density was obtained from Looyenga’s formula with an assumption of ice permittivity equal to 3.15 [14,15]:(3)$$\varepsilon_{snow}={\left(W\varepsilon_{w}^{\alpha }+I\varepsilon_{ice}^{\alpha }+A\varepsilon_{air}^{\alpha }\right)}^{\frac{1}{\alpha }}$$
where, *W*—water content, *I*—ice fraction, *A*—air in snow, *ε_w_*—water permittivity, *ε_ice_*—ice permittivity, *ε_air_*—air permittivity, and *α* = 0.3.

Snow density was calculated knowing that:(4)W+I+A=1,
and assuming that *W* = 0.

## 3. Results

### 3.1. Effects of Soil Moisture Upon Off-Road Vehicle Mobility

Selected results of the vehicle mobility tests are presented in Figure 5. Here, we have values of *VMI* for four successive months of the 2006 year. Red bars on the graphs indicate the need for using the 4 × 4 drive mode. The vehicle was always started in 2 × 4 mode and, if the vehicle became immobile, it was then switched to 4 × 4.

It is visible that the *VMI* for early spring months (March and April) reached higher values than in May or even June. Atmospheric precipitations that occurred in May and June with a great intensity (thunderstorms, convective rains) affected vehicle mobility more strongly than the effects of snow cover of thawing soil during early spring. Furthermore, the number of cases with the need to switch to the 4 × 4 mode was higher in late spring/early summer. An interesting observation is that the dynamics of the *VMI* value were considerably higher in June than in March.

Figure 6 presents the average values of *VMI* for the months of the entire season 2006–2007. Averaging was done arithmetically by adding the results for a given month then dividing by the number of tests. The values fell between 0.2 and 0.74. Similar to early expectations, the lowest mobility was observed in winter/spring and the highest in summer/autumn. Low traction in winter and spring was typical as a result of mainly low temperatures and less solar radiation, which led to lower evaporation and lower soil water flux due to freezing. Although cumulative rainfalls in Poland are the greatest in May–July, the effects of ambient temperature, wind, and solar radiation were more pronounced upon soil MC (Moisture Content) and mobility. Typically, a huge rainfall during a June thunderstorm weakened the traction to “no-go” conditions, but the soil rebuilt its mechanical strength very quickly and the mobility was acceptable or good.

It is also interesting how the dynamics of vehicle mobility changed in each month. These dynamics were significantly greater during spring/summer months as a result of the weather factors discussed above. Huge rainfalls in summer, together with high ambient temperatures, wind, and sun radiation, resulted in dynamical changes in mobility. More stable weather in autumn, with very few precipitation events, lower temperatures, and less sun radiation, caused the lower soil MC and the resulting traction reduction, changing over time and the values of mobility were medium for those seasons. In early spring, when freezing/thawing cycles were dominant, vehicle mobility was poor. Based on the measurements, the worst mobility in winter was observed in the last week of February (27–28 February).

Here, we noticed some drawbacks in the test procedure. We aimed to repeat the measurements every week, but the days for the tests were chosen at random. Due to the high dynamics of soil MC in spring/summer months, it was possible that the measurements were not representative. We performed some measurements in summer after heavy rainfalls, with results of the *VMI* near to zero, while the traction on the remaining days of a given week were quite good or almost very good. However, the winter 2006/2007 in Poland was dry and warm. There were very few days with freezing temperatures and very little snow fell that winter. The only snow fell on 25 January 2007, where there was 25 cm of fresh powder snow, which disappeared in the following days very quickly. We had not repeated the measurements the following winters, but by simple comparison, the winter of 2009–2010 was probably the strongest and snowiest in the last 30 years in Poland.

Summarizing the results, some trends can be observed:Spring—a tendency for mobility values to increase in the early spring months, then strong dynamics in mobility value, especially in June;Summer—high dynamics of mobility in early summer, then almost stable at approximately the 75% level;Autumn—a tendency for mobility values to decrease with no dynamic changes, with an average mobility of 30–40%; andWinter—a tendency for mobility values to decrease at the beginning, then remain almost stable, with rather low values. At the end of this season, in March, there was an increase in mobility value together with some dynamics.

One interesting result was the observed need for switching to 4 × 4 mode, which is depicted with red bars on the graphs. Summarizing the results for all months we have the following:Spring—8 instances of 4 × 4 switching in a total of 20 tests;Summer—2 instances of 4 × 4 switching in 16 tests;Autumn—8 instances of 4 × 4 switching in 15 tests;Winter—10 instances of 4 × 4 switching in 19 tests.

As it can be noticed, the most frequent need for the 4 × 4 was in winter, spring, and autumn. This conclusion was in agreement with the mobility results above.

### 3.2. Relationship Between Vehicle Mobility and Soil Moisture

In order to model the effect of soil moisture upon vehicle mobility, a correlation between loess soil moisture and the *VMI* was created. The results of soil MC measurements are included in Figure 7, and based on the data for the loess soil, we calculated the correlation, which is shown in Figure 8. It is visible that the function of the correlation is decreasing, but its course is not uniform in the whole range of the soil MC. Consequently, the lower the soil MC, the higher the *VMI*. For a soil MC within 17% and 22%, vehicle mobility was almost unchanged, constant at the level of approximately 0.5. For an MC outside of 22%, the curve fitting shows that the *VMI* drops to lower values, but this should be taken with due caution, since the R^2^ is low (0.506). Although the soil plastic or liquid limits were not determined at the time of measurements, the behavior of the *VMI*, in terms of the soil MC, could be explained as the effect of the fraction of sand in the soil composition. The radical drop in *VMI* at MC = 17% may have been caused by the loss of cohesion, which is typical for sands with increasing moisture.

### 3.3. Dynamics of Soil Moisture Content of A Grassy Runway

The results of the measurement of soil MC of a grassy runway are presented in graphical form in Figure 9, Figure 10, Figure 11, Figure 12 and Figure 13, and include average soil moisture content values along with trend lines as well as the error bars.

Figure 9 shows the averaged values of soil moisture measured every hour. The measurement was started after a very intense rainfall (about 30 L/m^2^). As one can see, the dynamics of soil moisture change is high, where, within 5 h, the humidity decreased from 36.62% to 32.84%. Such an intense change over time occurred despite the fact that the atmospheric conditions were not favorable: air temperature was about 18 °C, full cloudiness, and weak wind. However, it should be noted that the initial value of soil moisture was very high (36.62%, the highest absolute average value of humidity registered during the study), which initially caused the very intense movement of water in the soil. The course of averaged values is described by a logarithmic curve, which is characterized by a very good fit with the experimental results (R^2^ = 0.96).

The next figure (Figure 10) shows the course of average values of soil moisture on the second day after rainfall. The measurements were taken every 3 h. Meteorological conditions were similar to those on the previous day, and only the temperature was slightly higher at 19 °C. Additionally, there were break intervals during the day. It can be noticed that the dynamics of soil moisture change was slightly smaller than before. The logarithmic curve, which describes the course of measured values, is characterized by a much lower adjustment coefficient, R^2^ = 0.76, which indicates fluctuations in humidity during the day, probably caused by sunshine, and, thus, more intensive evaporation of water from soil and plants.

Figure 11 shows the selected results of measurements carried out on May 19–21. The measurements were carried out at various time intervals. In the discussed period, the weather was full cloud, and, during the day, the temperature was around 19 °C, while at night it dropped to about 14 °C. On the first day, May 19, at around 20:00 (UTC +2), there was a convective rain with an intensity of 20 L/m^2^, while on the second day, May 20, rain of a similar intensity fell in the evening. Additionally, on May 21, the grass was mown in the morning. The results are described by two curves, one logarithmic, which is very weak (R^2^ = 0.36), and the 5th-order polynomial. In the case of the polynomial, both the R^2^ coefficient and the shape of the curve, in relation to the points on the graph, are much better.

The course of changes in soil moisture on May 22–24 is shown in Figure 12. Atmospheric conditions on these days were variable: during the first day, it was initially warm, almost hot (23–30 °C), and sunny, then on the next day it cooled to about 18 °C with an overcast sky. On May 24, in the evening (18:00 UTC +2), there was a small convective rainfall. The significant dynamics of soil moisture in the analyzed period was caused by atmospheric factors. At the end of the analyzed period, the moisture content of the soil dropped to approximately 20%. Similarly, as before (Figure 11), a much better fit was obtained in the case of a polynomial curve, but this time a 4-order polynomial was used.

Figure 13 presents a situation where the average values of soil moisture increased as a result of irrigation of the soil by precipitation. After a period of stable weather from 25 to 27 May, a convective rainfall occurred in the morning hours of 28 May, but it was not very intense. At midnight, a very intense rainfall occurred (about 40 L/m^2^). This caused a significant increase in soil moisture, where it reached 31.2%. The analytical description of the MC course was carried out using a polynomial curve and the fit is very good, where R^2^ = 0.99.

At the Radawiec grassy airfield, the majority of flight operations are training flights of light aircraft and gliders. On the basis of practical observations, it was found that the optimal soil moisture, at which safe aerial operations are performed without significant impact on ground performance, was about 18–23%. The maximum acceptable humidity reached 30%, while, in these conditions, there was a deterioration of the aircraft’s performance on the ground, which was manifested by a longer ground roll on takeoff. The absolute maximum value of soil moisture recorded in the 2014 season was 42%, when air operations were suspended. The presented research showed that it was necessary to monitor the soil moisture of a grassy runway with a frequency of at least one measurement per hour. This was due to the observed dynamics of changes in humidity, especially in the case of heavy rainfall, high external temperature, and sunshine. It can be stated that, especially in the case of the analysis of changes in humidity over more than one day, the description of the course and the polynomial function of a 4- or 5-order was much more advantageous than the logarithmic curve.

One good summary of this kind of research would be to build a model that would take into account all these factors. So far, known models of soil moisture dynamics primarily take into account the movement of water in the soil and the uptake of water by plant roots (Anderson et al., 1983, [5]). However, there are no models that take into account weather factors such as wind, sunshine, and ambient air temperature together with soil MC in a dynamic way. One difficulty in doing so is the lack of methods and results of real-time research. A partial remedy for this situation is the proposal of a field sensor that provides data on the soil moisture of the grass aerodrome and the height of the grass.

### 3.4. Design Idea of a Field Sensor for Soil MC Remote Measurements

High dynamics of soil MC in a grassy airfield surface required frequent measurements in order to monitor all effects correctly. This, however, required a person, who operates the TDR, to be available almost all of the time during the measurement campaign. One good remedy for this is a remote measurement system, and such systems are available commercially. However, in addition to the soil MC, grass height is of importance for wheel performance and vehicle dynamics. Therefore, an idea of the authors was to design and develop a field-integrated sensor for soil MC and grass height that operates autonomously. The concept assumes that the sensor should be maintenance-free and that the gathered data could be collected from anywhere in the world, so there is no need for an operator to actually be present in the place of measurements. Another feature would be that the installation of the sensor on the airfield does not disturb ground traffic of airplanes, and that the mechanical design would be load resistant to the effect of wheeling or walking. The sensor should be easy to install or remove and could be left outdoor at ambient air temperatures not below −10 °C.

Generally, the field sensor design would consist of two sub-systems:-a TDR sensor for soil moisture content measurements, and-a grass high sensor.

The TDR sensor intended for use in the design is a commercially available unit with some modifications that ensure faultless operation under field conditions. These modifications would include:-adding a water- and dust-resistant case of a higher mechanical strength in order to ensure safe operation in the field;-installation of a high capacity, highly efficient power supply that would allow for long time operation; and-adding communication ports for linking with external devices and a miniature universal serial bus (USB) port.

The grass height (length) measurement module would use an optical sensor for detecting the presence of grass blades at a given height above the ground. The measurement method would be digital discrimination based on light permeability though the grass blades. Figure 14 depicts the design with details. For more information please refer to references [16,17].

The practical problem with a grassy airfield occurs in short runways, where an airplane could eventually not achieve liftoff, especially at high soil MC conditions. It is difficult to monitor and report runway conditions on seldom used, remote airfields. Incidents and accidents during takeoff or landing are reported in piloting or aviation magazines quite frequently [18,19,20,21,22,23]. Therefore, future research will focus on the research and development of the described field sensor and applications in the GARFIELD online information system, that is being developed by the authors [24,25].

### 3.5. Snow Density

Results from the grooming machine experiment, i.e., values for four different depths of snow, are included in Table 1. Together with TDR measurements, snow stress measurements were done with four single-axis pressure sensors. For more details, please refer to the research by Pytka, 2010 [13]. In the lower part of this Table, snow density data obtained in the airplane experiment are collected. The effect of the tractive element—wheel or ski—is evident: wheel loads caused almost 2.5-times higher snow compaction than ski loads.

## 4. Conclusions

In the paper research on soil moisture content for wheel and vehicle performance on natural, deformable terrain were performed.

Soil moisture content was measured by means of a handheld TDR meter in a mix of loess, sand, and gravel on a provisional road (test track), together with vehicle performance for the entire season 2006–2007, and a relationship between the soil MC and the vehicle mobility index (*VMI*) was derived from experimental data. It was assumed that the lower the soil MC, the higher the *VMI*, although the function describing this dependence was neither linear nor monotonic.

Soil MC was also determined in the surface of a grassy runway, together with observations of weather elements: precipitation, sun radiation, wind, and ambient air temperature. A family of correlations was created, together with a simple mathematical description. It was noticed that the soil moisture content on a grassy airfield changed very rapidly, especially due to convective rains (wetting), and high temperatures and sunshine (drying). A measurement frequency of about 1 per hour would be required for monitoring the grassy surface. Another important factor affecting changes in soil MC was grass height.

In the third part of this study, the TDR meter was used for indirect determination of snow density during two experiments, one with a grooming machine and another with a light airplane with ski gear on a snow covered runway.

The experiments presented in this study proved the high value and usefulness of the TDR measurement technology in terramechanical research. The method was sensitive enough for the factors that affected the off-road vehicle mobility. However, the need for the handheld TDR operation by a technician was disadvantageous because it was difficult to perform measurements in several places at a time. Therefore, one conceptual conclusion of the study was an idea for an integrated field sensor for soil moisture as well as grass blade length monitoring on a grassy airfield runway. The instrument would utilize the so called “internet of things” philosophy and would consist of a TDR sensor, an optical sensor, a microcomputer system, and a *GSM* communication module. The design should ensure the autonomous operation of the sensor with options for wired or wireless transmission of the measured data. One prospective practical use of such a sensor would be the GARFIELD online information system on the runway surface condition for grassy airfields.

## Figures and Tables

**Figure 1 sensors-19-02116-f001:**
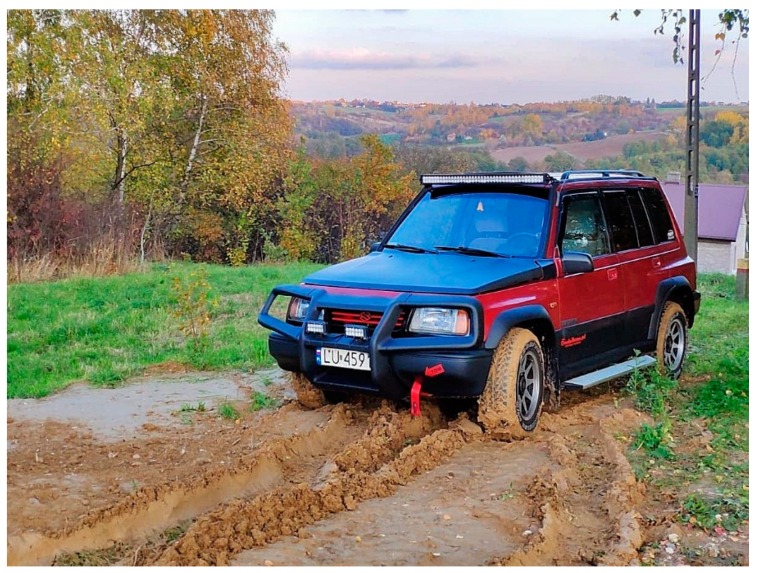
The Suzuki Vitara 1.6 16V used for vehicle mobility tests.

**Figure 2 sensors-19-02116-f002:**
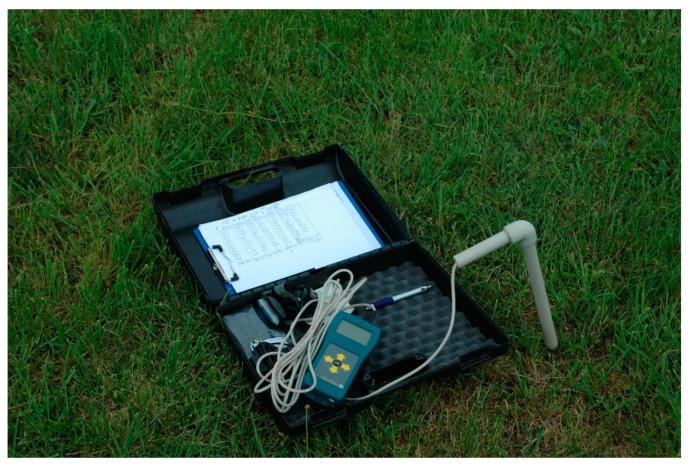
The handheld time-domain reflectometry (TDR) meter used for measurements of soil moisture content of grassy runway.

**Figure 3 sensors-19-02116-f003:**
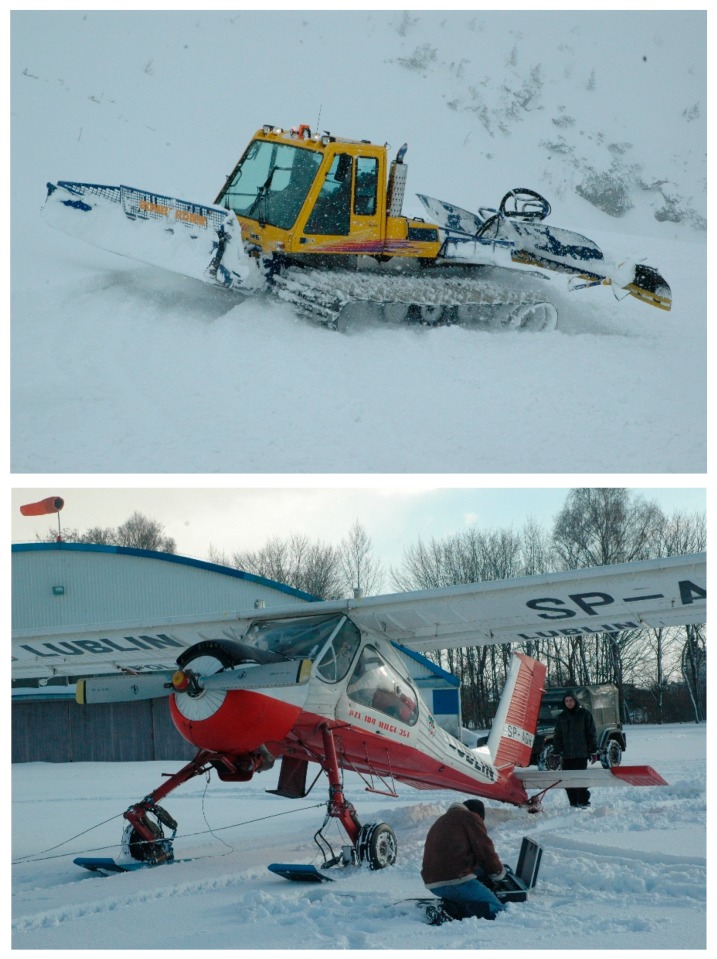
The two machines used in the snow measurements: the Bombardier snow grooming machine (upper photo) and the PZL 104 Wilga 35A airplane (lower photo).

**Figure 4 sensors-19-02116-f004:**
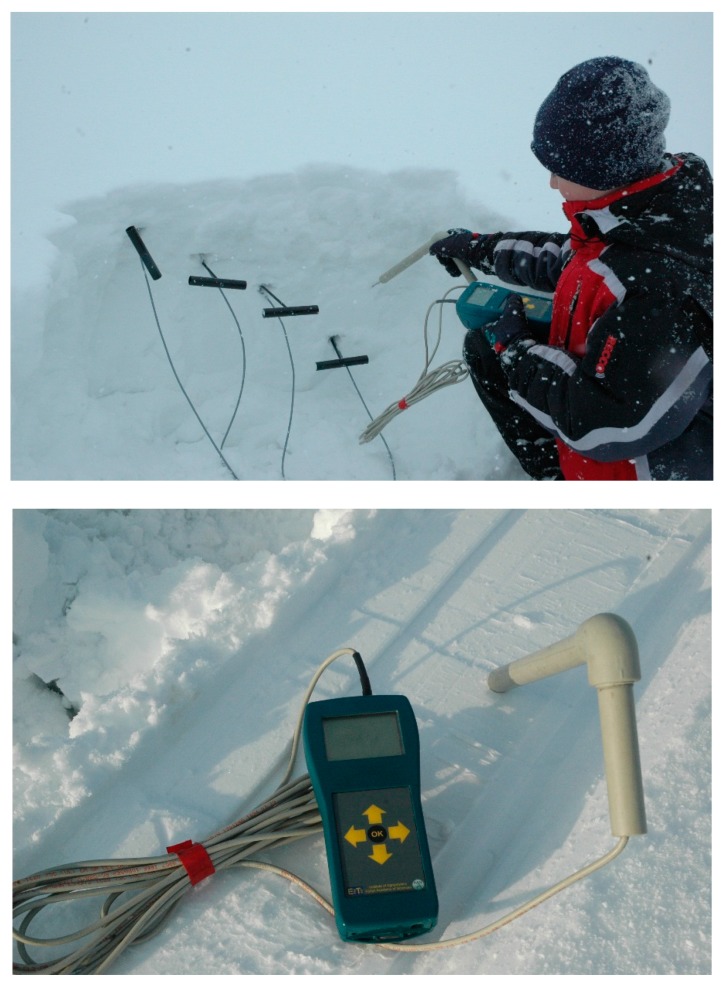
Determining the snow density with the TDR handheld meter: experiment with a grooming machine (upper photo) and with a light airplane (lower photo).

**Figure 5 sensors-19-02116-f005:**
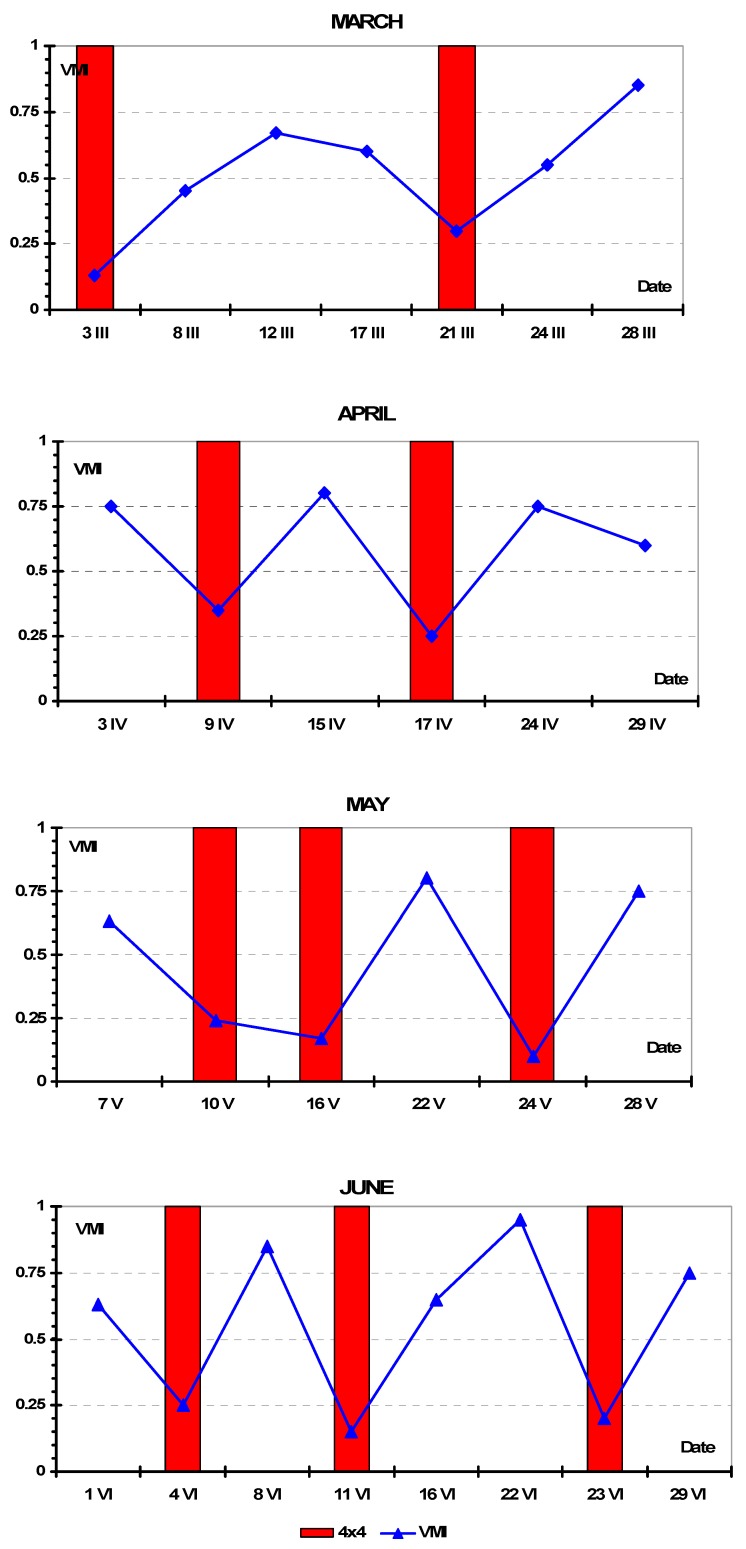
Results of the vehicle mobility tests performed during four months in the 2006 year.

**Figure 6 sensors-19-02116-f006:**
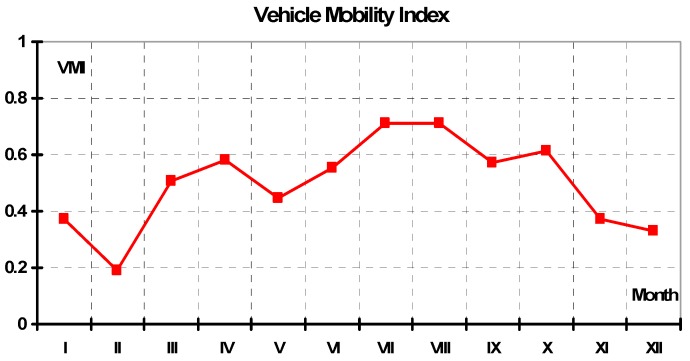
Average values of the vehicle mobility index (*VMI*) for the 2006–2007 season.

**Figure 7 sensors-19-02116-f007:**
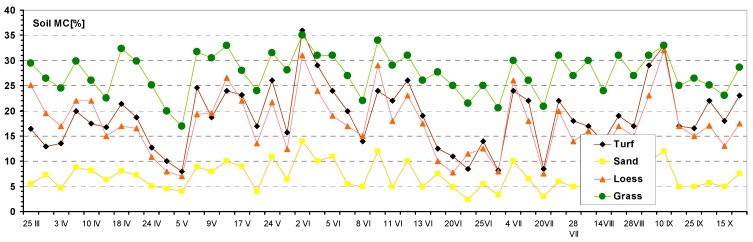
Results of the measurements of the soil MC for the four different soil surfaces during the warm months of the year 2007.

**Figure 8 sensors-19-02116-f008:**
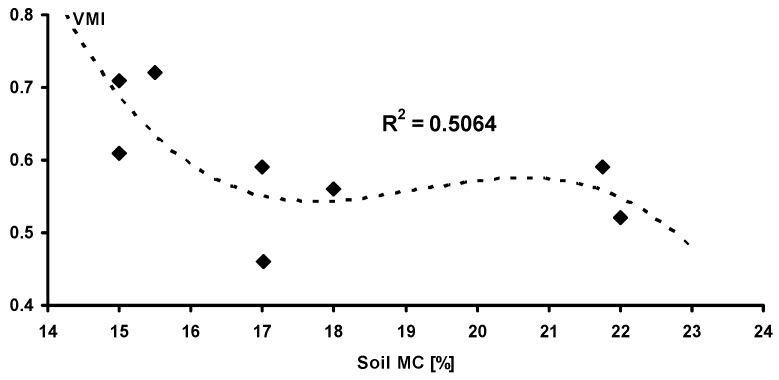
The relationship between the soil MC and the *VMI.*

**Figure 9 sensors-19-02116-f009:**
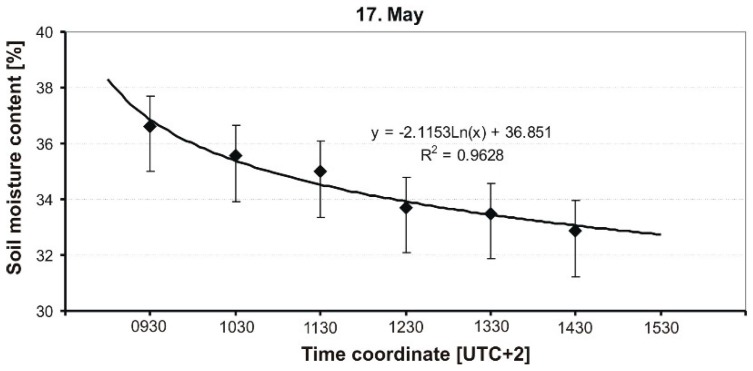
Dynamics of soil MC in a grassy runway on 17 May 2014.

**Figure 10 sensors-19-02116-f010:**
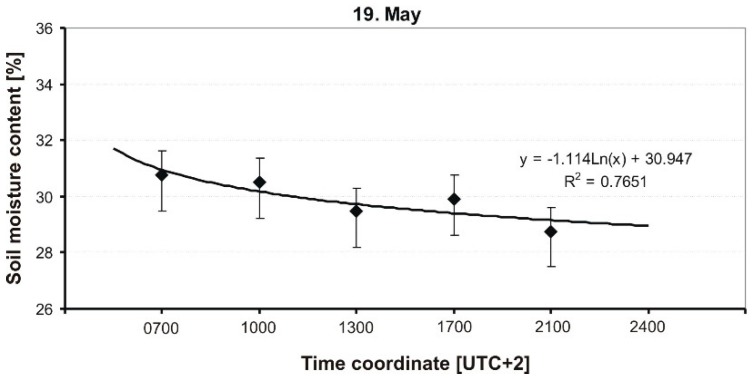
Dynamics of soil MC in a grassy runway on 19 May 2014.

**Figure 11 sensors-19-02116-f011:**
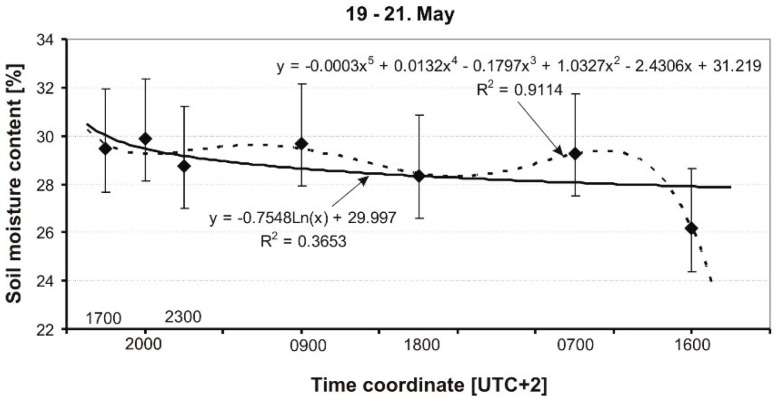
Dynamics of soil MC in a grassy runway on 19–21 May 2014.

**Figure 12 sensors-19-02116-f012:**
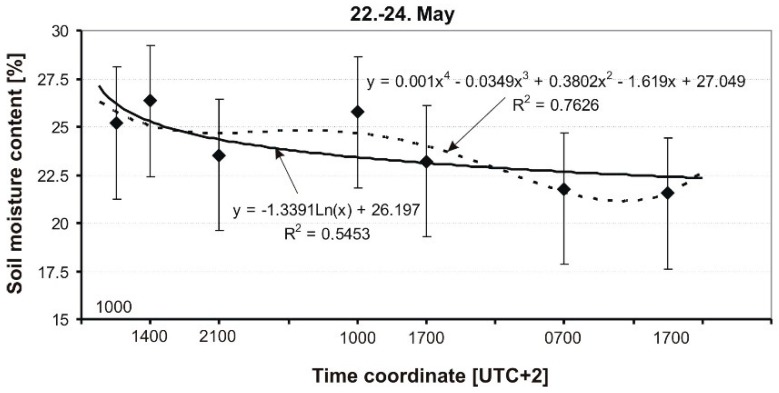
Dynamics of soil MC in a grassy runway on 22–24 May 2014.

**Figure 13 sensors-19-02116-f013:**
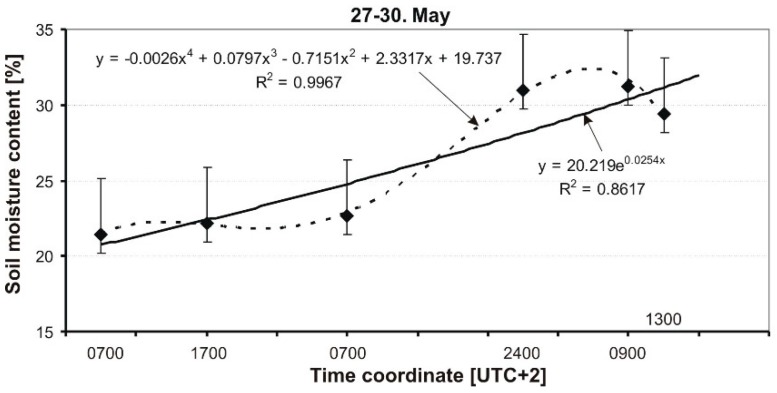
Dynamics of soil MC in a grassy runway on 27–30 May 2014.

**Figure 14 sensors-19-02116-f014:**
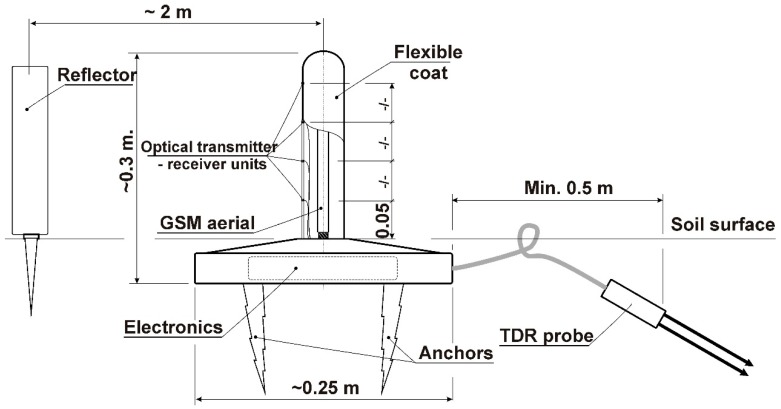
The integrated field sensor for soil moisture content and grass blade length monitoring.

**Table 1 sensors-19-02116-t001:** TDR readings and snow density data.

**Grooming Machine Experiment**
Depth	0–10 cm	10–20 cm	20–30 cm	30–40 cm
TDR, ε	1.07	1.17	1.25	1.46
Snow density [kg/m^3^]	129	311	447	778
**Airplane Experiment**
Snow	Undisturbed snow	Compacted snow-ski	Compacted snow-wheel	
TDR, ε	1.256 (0.027)	1.340 (0.015)	3.318 (0.363)	
Snow density [kg/m^3^]	447	612	1309

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
