# Peer review of "Application of the TDR Soil Moisture Sensor for Terramechanical Research†"

_sensors, 2019, doi:10.3390/s19092116_

Round 1
Reviewer 1 Report
The manuscript is well written and interesting.
The line 81 carries the expression "Vehicle Mobility Indicator" but other parts explain the VMI as vehicle mobility index (as in line 379). Unified use of term is recommended.
For the sentence at line 247-248 for Fig. 8, can we say that the VMI drops at soil MC larger than (i.e. outside of) 22% by the result of curve fitting with R^2 = 0.506? Additional soil physical data such as plastic limit and liquid limit might be helpful to understand the behavior of VMI in terms of soil MC. This might also explain the low VMI around soil MC of 17%.
At line 50, the word "compare" seems to be "compared."
Fig. 9 at line 270 seems to be Fig. 10.
Author Response
The manuscript is well written and interesting.
- Thank you very much.
The line 81 carries the expression "Vehicle Mobility Indicator" but other parts explain the VMI as vehicle mobility index (as in line 379). Unified use of term is recommended.
- We’ve unified the term (VMI = Vehicle Mobility Index);
For the sentence at line 247-248 for Fig. 8, can we say that the VMI drops at soil MC larger than (i.e. outside of) 22% by the result of curve fitting with R^2 = 0.506? Additional soil physical data such as plastic limit and liquid limit might be helpful to understand the behavior of VMI in terms of soil MC. This might also explain the low VMI around soil MC of 17%.
- We’ve added the following text (lines 244 – 249):
”For the MC outside of 22%, the curve fitting shows that the VMI drops to lower values, but this should be taken with due caution, since the R2 is low (0,506). Although the soil plastic or liquid limits were not determined at the time of measurements, this behavior of the VMI in terms of the soil MC can be explained as the effect of sand fraction in the soil composition. The radical drop of soil MC of 17 may be caused by the loss of cohesion, typical for sands at increasing moisture.”
At line 50, the word "compare" seems to be "compared."
- Agreed, we’ve corrected.
Fig. 9 at line 270 seems to be Fig. 10.
- Agreed.
Thank you very much for your valuable comments. Please note that the ”colour code” for our reply to your comments in the MS WORD file is red letters with yellow background.
Reviewer 2 Report
I have made comments on the pdf file attached.
Overall quality of the article is good and to be accepted with minor revision.

Author Response
Thank you for your valuable comments.
All the comments have been addressed in the PDF file.
Please note that the ”colour code” for our corrections based on your comments in the MS WORD file is blue letters with yellow background.
Thank you once again.

This manuscript is a resubmission of an earlier submission. The following is a list of the peer review reports and author responses from that submission.